# Common Marmoset Cell Lines and Their Applications in Biomedical Research

**DOI:** 10.3390/cells12162020

**Published:** 2023-08-08

**Authors:** Ekaterina Bayurova, Alla Zhitkevich, Daria Avdoshina, Natalya Kupriyanova, Yuliya Kolyako, Dmitry Kostyushev, Ilya Gordeychuk

**Affiliations:** 1Chumakov Federal Scientific Center for Research and Development of Immune-and-Biological Products of Russian Academy of Sciences, 108819 Moscow, Russia; bayurova_eo@chumakovs.su (E.B.); zhitkevich_as@chumakovs.su (A.Z.); avdoshina_dv@chumakovs.su (D.A.); kupriyanova_ns@chumakovs.su (N.K.); kolyako_jv@chumakovs.su (Y.K.); 2Institute for Translational Medicine and Biotechnology, Sechenov University, 117418 Moscow, Russia; 3Martsinovsky Institute of Medical Parasitology, Tropical and Vector-Borne Diseases, Sechenov University, 119435 Moscow, Russia; dkostushev@gmail.com; 4Scientific Center for Genetics and Life Sciences, Division of Biotechnology, Sirius University of Science and Technology, 354340 Sochi, Russia

**Keywords:** common marmosets, *Callithrix jacchus*, cell lines, iPS, embryonic stem cells, animal models, lymphoblastoid cell lines, primary cells

## Abstract

Common marmosets (*Callithrix jacchus*; CMs) are small New World primates widely used in biomedical research. Early stages of such research often include in vitro experiments which require standardized and well-characterized CM cell cultures derived from different tissues. Despite the long history of laboratory work with CMs and high translational potential of such studies, the number of available standardized, well-defined, stable, and validated CM cell lines is still small. While primary cells and immortalized cell lines are mostly used for the studies of infectious diseases, biochemical research, and targeted gene therapy, the main current applications of CM embryonic stem cells and induced pluripotent stem cells are regenerative medicine, stem cell research, generation of transgenic CMs, transplantology, cell therapy, reproductive physiology, oncology, and neurodegenerative diseases. In this review we summarize the data on the main advantages, drawbacks and research applications of CM cell lines published to date including primary cells, immortalized cell lines, lymphoblastoid cell lines, embryonic stem cells, and induced pluripotent stem cells.

## 1. Introduction

Common marmosets (*Callithrix jacchus*; CMs) are small New World primates that have been increasingly used in many biomedical research areas including infectious diseases, oncology, autoimmune diseases, age-related disorders, neuroscience, reproductive biology, stem cell research, and toxicology [1,2,3,4,5,6].

CMs are naturally susceptible to multiple human viral, protozoan and bacterial pathogens [7] including Yellow fever [8], Epstein-Barr virus (EBV) and other Herpesviruses, hepatitis A virus [9], Junin virus, malaria, measles, hepatitis E virus [10]. With multiple methods of assessment of humoral and cell-based immune response developed in recent years, this makes CMs an optimal model for preclinical studies of safety and efficacy of novel regimens of prophylaxis and treatment of infectious diseases, including vaccines, immunotherapeutics and cell-based therapies [11]. The main applications of CMs in biomedical research are summarized in Figure 1A.

Substantial advantages of using CMs in biomedical research include their small size (300–500 g), relative ease of maintenance and breeding in captivity, phylogenetic closeness to humans, and short gestation period combined with fast sexual maturation [3,7]. Many commercially available monoclonal antibodies used for differentiation of human cell populations by flow cytometry cross-react with the corresponding CM antigens [12,13,14,15], which makes it possible to apply the well-established research methods commonly used in human immunological studies.

Unlike most other primate species, marmoset litter predominantly contains two or more infants [2], which enables experimental design with matched twins in experimental groups. Moreover, members of the Callitrichidae family (marmosets and tamarins) are uniquely capable of producing polyzygotic twins with hematopoetic chimerism of the infants from the same litter. Chorionic fusion of the placentas during gestation leads to formation of a single chorion with anastomoses connecting the embryos. This fusion process allows for the exchange of hematopoetic progenitor cells via blood flow between the twins [16].

Another characteristic feature of the CMs is minimal diversity at both major histocompatibility (MHC) Classes I and II loci [17,18,19]. One consequence of this reduced variation is that callitrichids are much more tolerant to transplantation [20,21], with xenografts between species within the family surviving an average of three weeks, and allografts within species often lasting months [22,23], potentially making it possible to perform allotransplantation between CMs in the absence of immunosuppressive agents [24].

Still, however, the use of all of the abovementioned valuable properties of CMs in biomedical research is hampered by the low number of available cell cultures (Figure 1B). Therefore, development of standardized, well-defined, stable, and validated CM cell lines, as well as effective cell immortalization protocols, opens access to new modelling systems for a wide spectrum of human diseases in small non-human primates. In this review we summarize the data about all currently available CM cell lines, and the main areas of their application.

## 2. Primary CM Cells

Different primary CM cell cultures are used in experiments without any modifications. The first case was reported in 1978 by Falk et al. who used monolayer primary marmoset kidney cells for propagation of *Herpesvirus ateles* and as a feeder layer for marmoset lymphocytes during the development of lymphoblastoid cell lines (described below) [25]. Primary CM fibroblasts of different origin were widely used by researchers due to the ease of obtaining. Primary skin fibroblasts were used in research on reactive oxygen species production, and experimental testing of oxidative stress hypothesis of aging [26]. Primary skin fibroblasts were obtained from skin biopsy samples and cultured for quantitative assessment of chimerism in marmosets and tamarins [27], and lately cultured fibroblasts obtained by trypsinization of small pieces of tissue from the ear were used to verify transgene integration and expression in genetically modified CMs [28]. In a work by Pogozhykh et al. isolation and systematic study of multipotent stromal cells (MSCs) derived from human and CM amnion and bone marrow was described. The studied human and CM samples share many similar features such as most MSC markers and reduced MHC class I expression in amnion cells vs. bone marrow [29]. Cultured CM bone marrow progenitor cells can be stimulated by human cytokines and differentiated into adipocytes, osteocytes and chondrocytes in vitro [30,31].

The methods of obtaining primary CM cells were described in several studies. For example, dendritic cells were isolated from blood, spleen and bone marrow in the amounts sufficient for use in preclinical studies of cell therapy for central nervous system diseases and cancer [32]. Cell cultures of CM reproductive system were also obtained, namely peritubular cells [33] and ovarian cells [34]. Lately, Jang et al. described isolation and characterization of the primary retinal pigment epithelial cells, where the quality of obtained culture correlated with donor age [35]. The authors emphasize that while CMs are widely used for modeling of age-related macular degeneration, no retinal pigment epithelial cells had been previously described as an alternative to in vivo experiments.

Due to natural susceptibility to different hepatotropic viruses such as hepatitis E virus [10], and other viruses causing liver pathology such as lymphocytic choriomeningitis virus (causative agent of Callitrichid hepatitis) [36] and Lassa virus [37], the development of CM hepatocytes culture is an important task. Primary CM hepatocyte cultures were obtained in 1991 by Stephensen et al. Cells were characterized by stable expression of apolipoproteins Al and E for over 66 days in culture and were used to characterize Callitrichid Hepatitis associated virus as the etiologic agent of Callitrichid hepatitis [38].

GB virus-B (GBV-B) causing acute self-limiting infection in a variety of New world monkeys is closely related to hepatitis C virus, and antiviral therapies effective against GBV-B could be effective against HCV [39,40,41]. In 2000 Beames et al. described the development of primary tamarin hepatocytes culture susceptible to GBV-B as surrogate model for HCV infection [42]. Later several groups used this method for the purpose of obtaining primary CM hepatocytes. The resulting cell cultures were used to study GBV-B infection and the efficacy of antiviral drugs [41,43,44]. Additionally to the GBV-B surrogate models of HCV infection, in 2006 Martyn et al. obtained primary marmoset hepatocytes and transduced them with recombinant baculovirus vector encoding E1 and E2 envelope proteins of HCV [45]. This experiment provides an in vitro model system for HCV gene expression studies and confirms the possibility of highly effective transduction of primary marmoset cells with baculovirus vectors.

In 2007 Chin et al. used primary hepatocytes of CMs for HBV studies. The authors used adenovirus system to deliver replication competent HBV genome into primary cells to dissect mechanisms of viral pathogenesis [46]. Importantly, the authors demonstrated that CM hepatocytes are permissive to HBV replication, while it is not true for murine hepatocytes [47].

Thus, primary CM cells could be used for modeling of viral infections and as a valuable tool for reproduction research and other in vivo studies. However, in general the use of primary cells is limited by relatively short life span in culture and variability between batches, which demands the development of effective immortalization techniques.

## 3. Immortalization

Primary cell cultures have certain limitations including poor standardization of the experiments due to short culturing period, and requiring for repeated cells isolation for every experiment. Since after a certain number of mitotic divisions most cells enter the phase of replicative senescence [48], the applicability of terminally differentiated somatic cells in vitro for research or therapeutic use is limited. Progressive shortening of telomere ends [49] leads to damage of genomic DNA, and activation of p53/p21-mediated cell cycle control pathways [50,51,52] resulting in cell growth arrest (M1 senescence) and, if cells continue to proliferate, in mass cell death (M2 senescence) [52]. Blocking of cell cycle checkpoint pathways and restoration of telomerase activity are key factors for cell immortalization.

The first successful attempts to immortalize CM cells were made in 2002 in ovarian granulosa cells and theca cells by transfection with the classical oncogene, simian virus 40 large T antigen (SV40LT) [53]. The generated cell lines retained several tissue-specific features (e.g., hormone responsiveness and specific enzyme expression) providing an extremely useful experimental test system for biomedical studies of reproductive physiology, in particular, the process of luteinization.

Recently, several immortalized cell lines were developed. First, Petkov et al. showed that marmoset fibroblasts were successfully immortalized with transposon-integrated transgenic human telomerase reverse transcriptase (hTERT) and expanded in vitro for over 500 population doublings, while the wild-type fibroblasts only reached a maximum of 46 doublings. The immortalized cells exhibited differences in morphology as compared to the control fibroblasts, and transcriptome analysis revealed changes in gene expression patterns, but one sub-clonal line with normal karyotype was established. The results of this study were an important step towards the development and optimization of methods for the production of immortalized CM cells [54].

Guo et al. produced immortalized marmoset hepatic progenitor cells (MHPCs) by lentivirus-mediated transfer of the SV40LT gene into fetal liver polygonal cells. These cells possess hepatic progenitor cell-specific gene expression profiles with potential to differentiate into both hepatocytic and cholangiocytic lineages in vitro and in vivo and can be genetically modified for use in disease modeling, development of treatment regimens, and allotransplantation therapy for liver diseases [55].

Recently, several groups reported immortalization of CM fibroblasts. Orimoto et al. used piggyBac transposition of the mutated form of cyclin-dependent kinase 4, Cyclin D1 and hTERT. The generated immortalized CMs fibroblasts (K4DT cell line) exhibited telomerase activity and an accelerated cell proliferation rate [56]. Jeong et al. reported that CRISPR-Cas9-mediated targeting of the p53 gene or CDKN2A locus is effective for immortalizing primary CM skin fibroblasts. It was shown that CDKN2A gene knockout could exert a comparable effect to introducing the CDK4R24C transgene which predisposes humans to hereditary melanoma and abolishes the ability of CDK4 protein to bind to p16INK4A protein [57,58]. CM cells immortalized by CDKN2A retain functional p53 as opposed to immortalization with SV40LT [59].

The immortalized cell lines may become a useful tool for future studies on disease modeling and targeted gene therapy. Such cell lines can be used for reproducible in vitro studies in virology, drug metabolism, and oncology with a high potential of translation into in vivo studies.

## 4. Lymphoblastoid Cell Lines

CMs are a well-known animal model for Epstein-Barr virus (EBV) infection. In contrast to the other New World monkey species, the cotton-top tamarin (Saguinus oedipus), EBV causes persistent infection in CMs without tumor formation [60]. This led to mass development of EBV-immortalized lymphoblastoid cell lines. In 1976 Desgranges et al. developed the first two CM lymphoblastoid cell lines: M81 and M72. Leukocytes were transformed by infection with EBV strain HKLY-28 derived from nasopharyngeal carcinoma. Generated lymphoblastoid cell lines M81 and M72 support EBV infection and produce EBV viral particles [61]. Later Wedderburn et al. used EBV produced by M81 cells to infect several CMs and obtained a panel of EBV-immortalized lymphoblastoid cell lines, namely M232, M242, M245 and M287 [62]. EBV-positive lymphoblastoid cell lines were mostly used for EBV viral particles production [62,63,64,65,66,67,68]. However, several works based on these lines were focused on EBV pathology. For example, Hotchin et al. used M245 B cells immortalized with M81-derived EBV to define tumorigenicity of lymphoblastoid cells expressing a constitutively activated *c-myc* gene [69]. Interestingly, the authors were able to inoculate immortalized B cells of one animal into its immunocompetent siblings for tumorigenicity assay, as CMs from the same litter are hematologically chimeric as a result of anastomoses between placentas and are thus tolerant to each other’s haemopoietic cells. The authors concluded that expression of c-myc is not sufficient to induce tumorigenic phenotype and at the same time observed immune clearance of cells expressing EBV antigens in inoculated animals. Also, using M81-derived EBV Wedderburn et al. have shown that EBV infection of marmosets could be long-term (up to 10 years) and asymptomatic as in humans [60]. M81 cells in combination with cotton-top tamarin and human cell lines were also used to define intracellular localization of EBV DNA during infection [70].

The other panel of lymphoblastoid cell lines was generated by immortalization following *Herpesvirus ateles* [25] or *H. saimiri* infection [71,72,73,74,75,76,77,78]. These cell lines were widely used as models of herpesvirus infection. Generation of herpesvirus-immortalized cell lines revealed restricted viral lymphotropism to the population of lymphocytes with NK cell function and phenotypic markers of both T cells and NK cells [71]. Using infection of CM lymphocytes with *H. saimiri* Letvin et al. made first attempts to dissect oncogenic mechanisms of herpesvirus infection. They established left-end L-DNA region of herpesvirus genome as a factor of viral oncogenicity [72]. CM lymphoblastoid cell lines were used for the first description of posttranscriptional regulation of herpesvirus U-rich microRNA [73] and its role in oncogenicity as well as oncogenic properties of viral proteins [75,76] was later studied [74,79].

The possibility to transform lymphocytes of CMs with both EBV and herpesviruses provided a model for studying functional properties of both transformed B and T cells from the same species of non-human primates [25]. However, the absence of easily accessible stable lines of herpesvirus-immortalized lymphoblastoid CM cells led to their relatively low use. Of note, herpesvirus-transformed cell line B95-8 originating from cotton-top tamarins is commercially available and cited in a plethora of articles.

It is important to note that CM primary, immortalized, and lymphoblastoid cell lines could be obtained and cultured using standard media and supplements used for human cells.

## 5. Common Marmoset Embryonic Stem Cells

Embryonic stem (ES) cells are pluripotent stem cells derived from the inner cell mass of blastocysts, primordial germ cells, teratocarcinomas and male germ cells that are capable of differentiating into all three germ layers [80]. Relatively early sexual maturation age and availability of techniques for ovulation control and synchronization between donor and recipient allowing to obtain blastocysts for transplantation monthly makes CMs an excellent primate species for the generation of transgenic and knockout animal models of human diseases as well as a suitable non-human primate model for research purposes in the fields of regenerative medicine and degenerative neural diseases [24]

First isolation of marmoset ES cells was reported in 1996 by Thomson et al. [81]. In this work eight cell lines were isolated and two of them, namely Cj11 and Cj62, were cultured continuously for over one year and remained undifferentiated and euploid. Several characteristics of these cell lines indicated that they belonged to ES cells, namely rapid proliferation for at least 18 months in continuous culture maintaining a normal karyotype, the expression of a combination of cell surface markers of monkey and human ES cells (SSEA-3, SSEA-4, TRA-1-60, TRA-1-81 and alkaline phosphatase), and the potential to differentiate to both endoderm and trophoblast. Later by developing an embryo collection system that ensures a stable supply of CM embryos for future production of transgenic or gene knockout marmosets Sasaki et al. have established three novel CM embryonic stem cell (CMESC) lines: CMESC20, CMESC40, and CMESC52 which showed three germ layer differentiation capacity, pluripotency, and expression of stage-specific embryonic antigens. It has been shown that these CMESC lines can differentiate into functional cardiomyocytes [82], neurons and glia in vitro and induce formation of teratomas that consisted of cartilage, adipose tissue, skeletal muscle, a bronchus-like structure, keratinizing squamous epidermis, epidermis, and CD31-positive vascular endothelial cells upon injection into immunodeficient mice [83]. The CMESC-derived neural stem/progenitor cells developed into neurons, astrocytes and oligodendrocytes in vivo upon allogenic transplantation providing a valuable preclinical model for the therapy of spinal cord injury [84]. In 2009 Muller et al. created and characterized a new line (cjes001) of ES cells from the CM that could be cultivated for up to passage 84 [85]. Later on it was shown that CM ES cells could be obtained not only from blastocyst but from natural morula stage preimplantation embryos [86]. In 2020 Yoshimatsu et al. reported the generation of a male marmoset embryonic stem cell line DSY127-BV8VT1 harboring BLIMP1 and DDX4 double reporters which are specifically expressed in germ cells and play pivotal roles in the development of the germ cell linage. This ES cell line will be a useful tool for investigating male gametogenesis in non-human primates [87]. In another study the CRISPR-Cas9 system was used to generate a CMES40-OC cell line carrying a novel OCT4 (POU class 5 homeobox 1) knock-in reporter that will be valuable for investigation of primed/naïve pluripotency and germ cell fate [88].

ES cells listed above were later differentiated into immunosuppressive macrophages [89], osteoblasts [90], retinal pigmented epithelium [91] or modified by introduction of recombinase-mediated cassette exchange [92]. Using CM40 [83] and Cj11 [81] cell lines Nii et al. showed that CM ES cells in terms of their morphology, gene expression, and growth factor dependency for self-renewal [93] and, therefore, present a reliable model for research in regenerative medicine. Later, the authors reported a novel and efficient method for differentiating CM ES cells into hematopoietic cells by transiently inhibiting the phosphoinositide 3-kinase (PI3K)-Protein kinase B pathway [94].

Recently, Aravalli et al. have carried out efficient hepatic differentiation of marmoset embryonic stem cells (ESCs) into functional hepatocyte-like cells (HLC) and demonstrated that the generated HLCs possessed specific characteristics similar to those of primary human hepatocytes. HLCs might be an optimal model for research in cell therapy of human liver diseases including hepatotropic virus infections, while the expression of CYP genes involved in the breakdown of various toxic molecules and chemicals also makes HLCs suitable for research of drug metabolism [95]. The potential for neural differentiation has been reported by Shimada et al. [96]. Two CM ES cell lines were used for the successful establishment of a highly efficient knock-in method for marmoset ES cells using CRISPR-Cas9 system for directed repair of DNA double-strand breaks on the example of the proteolipid protein 1 (PLP1) and forkhead box protein P2 (FOXP2) genes which are promising candidates for modification in the CM model. This method dramatically increased the number of colonies that survived positive selection and enabled bi-allelic homologous recombination [97]. These works could provide a basis for further application of ES cells of CMs beyond regenerative medicine.

Recently, temporal control of the tamoxifen-regulated Cre driver was demonstrated using a novel CM ES cell line, ActiCre-B1. Time-controlled genetic modification makes it possible to analyze phenotypes associated with embryonic lethality by knockout of functionally important genes. The new ActiCre-B1 cell line will provide a valuable research platform for studying gene knockout in non-human primate pluripotent stem cells [98].

Almost all marmoset ES cells have to date depended on mouse embryonic fibroblasts (MEFs) as feeder cells. However, the possibility of cultivating ESCs on Matrigel in conditions without feeder support (in feeder-free environments) has been described [95]. Therefore, contamination with the mouse cell components is an important issue for molecular or cellular biological analyses [99,100]. Furthermore, in contrast to macaque ES cells, marmoset ES cells derived from naturally-fertilized embryos were usually collected by uterine flushing procedure [81,83,86]. In a recent study by Kishimoto et al. 17 new marmoset ES cell lines were established from both naturally-fertilized and in vitro fertilized embryos under both feeder and feeder-free conditions. Furthermore, six of the 17 ESC lines carried male karyotype [101]. Male ES cell lines will be more useful for in vitro study of differentiation to sperm and for effective production of genetically modified marmoset models such as conditional knock out/in of the Y chromosome-specific genes, which has not yet been achieved in the CMs.

In a recent article by Kodera et al. cortical organoids and ganglionic eminence organoids were induced from cjESCs and fused to generate cerebral assembloids [102]. Due to the wide use of CM in neurobiological research, the marmoset assembloid system will provide a vitally needed in vitro platform for non-human primate neurobiology.

## 6. Marmoset Induced Pluripotent Cells

In 2006 Takahashi and Yamanaka proved that the factors that play important roles in the maintenance of ES cell identity also play pivotal roles in the induction of pluripotency in somatic cells [103]. In their groundbreaking work, four factors (namely Oct3/4, Sox2, c-Myc and Klf4), now known as Yamanaka factors, were sufficient to induce pluripotent stem cells from mouse embryonic or adult fibroblasts. Delivery of reprogramming factors with retroviral vectors raised safety concerns such as insertional inactivation of tumor suppressor genes and/or insertional activation of oncogenes, and other risks associated with constitutive expression of the reprogramming factors. This led to the use of transient, integration-free methods of delivering the reprogramming factors such as delivery/transient transfection with Sendai virus, adenovirus, episomal plasmids, minicircle plasmids, mini-intronic plasmids, PiggyBac transposons, synthetic modified mRNAs or miRNAs. Episomal plasmids and Sendai virus infection have been the preferred methods of choice for deriving clinical grade induced pluripotent stem cells (iPSCs) [104].

CM iPSCs were generated using retroviral transduction [105], non-integrative episomal vectors [106,107], PiggyBac system [108], and by adding chemical compounds, reprogramming factors and interferon suppressors to a conventional RNA transfection method [109].

The first work on the generation of marmoset iPSCs was published in 2010 by Wu et al. Several lines of iPSCs from newborn marmoset fibroblasts were obtained using retroviral transduction with human Oct4, Sox2, Klf4 and c-Myc. Generated cells fulfil critical criteria for successful reprogramming: they exhibit normal karyotype, are alkaline phosphatase positive, express high levels of NANOG, OCT4 and SOX2 mRNAs, are immunoreactive for Oct4 in the nucleus and TRA-1-81 and SSEA-4 in the plasma membrane, and when implanted into immunodeficient mice, produce teratomas that have derivatives of all three germ layers [105]. These experiments provide proof of principle that iPSC technology can be adapted for use in the marmoset as a future model of autologous cell therapy. Later efficient protocol for the directed neural differentiation of these pluripotent cells for experimental cell therapy was developed [110]. In the same year iPSCs derived from fetal liver of CM via the retrovirus-mediated introduction of six human transcription factors (Oct3/4, Sox2, Klf4, c-Myc, Nanog and Lin28 [111]) was reported.

Another approach was presented by Yoshimatsu et al. using EBV-based episomal vector system that provides persistent transgene expression which is advantageous for the efficient production of transgene-induced pluripotent stem cells without viral transduction. The introduction of additional reprogramming factors (KDM4D, GLIS1 and a p53 shRNA) into the episomal vector system allowed to create an iPSC line from somatic fibroblasts of a neonatal CM that showed standard pluripotency characteristics and could be a useful resource for stem cell research using non-human primates [112].

Due to wide use of marmosets for modeling of neurodegenerative diseases, several works focused on generation of iPSCs and their further differentiation into neuronal cells [107,113]. The data presented by Vermilyea et al. demonstrate that iPSCs can be efficiently differentiated to neurons as well as patterned to have a floor plate-derived midbrain dopaminergic phenotype that can be used for in vitro experiments on neural differentiation and to support Parkinson’s disease-related studies [114]. Besides, the same group reported successful use of CRISPR/Cas9 to introduce the LRRK2 G2019S mutation associated with 1–3% of Parkinson’s disease cases worldwide into marmoset ES cells and iPSC. It was found that, similar to humans, marmoset LRRK2 G2019S resulted in elevated kinase activity, increased intracellular reactive oxygen species, decreased neuronal viability and reduced neurite complexity. These results demonstrate the feasibility of inducing monogenic mutations in CMs and support the use of this species for generating a novel genetic-based model of Parkinson’s disease expressing physiological levels of LRRK2 G2019S [115].

Yamaguchi et al. generated CM iPSCs by lentiviral transduction of reprogramming factors including POU5F1 (also known as OCT3/4), SOX2, KLF4 and c-MYC into CM fibroblasts. The obtained cells showed an abnormal karyotype denoted as 46, X, del(4q),+mar and formed tumors similar to human dysgerminoma in severe combined immunodeficiency mice. CM dysgerminoma-like tumors were highly sensitive to DNA-damaging agents, irradiation, and fibroblast growth factor receptor inhibitor, and their growth was dependent on c-MYC expression [116]. It is known that iPSCs can form both teratomas and malignant tumors such as neuroblastoma and follicular carcinoma if transplanted in their undifferentiated pluripotent state in vivo [117], and tumorigenicity still is one of the concerns for iPSCs therapy [104]. Obtained CM dysgerminoma-like tumors could serve as a model for the development of strategies to deal with tumors unexpectedly formed in patients treated with iPSC-based therapies.

As mentioned above, lentiviral transduction with reprograming factors raised safety concerns which resulted in the development of integration-free methods of delivering the reprogramming factors [104]. These methods were also applied for generation of marmoset iPSCs. For example, Wiedemann et al. reprogrammed bone marrow–derived MSCs of adult CMs in the presence of TAV, SB431542, PD0325901 and ascorbic acid via a novel, excisable lentiviral spleen focus-forming virus (SFFV)-driven quad-cistronic vector system (OCT3/4, KLF4, SOX2, C-MYC) [118]. Later, Debowski et al. described the generation of eight pluripotent iPS cell lines from CM postnatal skin fibroblasts using a six-factor-in-one-construct piggyBac system including KLF4, c-MYC, SOX2, OCT4, NANOG, and the RNA binding protein LIN28 which were characterized in comparison with ES cells. The cells were stable over time proving that marmoset iPS cells generated using the piggyBac system may serve as a model for testing treatment regimens for cell and tissue degenerative diseases such as myocardial degeneration or neurodegenerative diseases (e.g., Parkinson’s disease) [108].

It is worth mentioning that marmoset and human reprogramming factors are very similar [105], so that human reprograming factors could be used for generating marmoset iPSCs. CM iPSCs as well as ESCs could be cultured in standard conditions required for human ESCs using serum-free media (knock-out media) supplemented with serum substitutes. In the most recent study CM iPSCs feeder-free culturing environment using Matrigel was described [114] demonstrating that culturing of CM cells is similar to that of human cells. Recently numerous studies developing CM ES cells and iPSCs and their potential use for regenerative medicine, pre-clinical trials and Parkinson’s disease therapy were published. However, their use in daily laboratory practice is still limited. In a review by Aravalli et al. [119] a list of advantages and potential applications of CM ES cells and iPSCs in liver disease modeling, tissue engineering and drug metabolism was described, but no articles describing the use of CM cells were mentioned. Importantly, the use of CM ES cells and iPSCs for modeling viral infections is also limited.

## 7. Conclusions

Biological characteristics of CMs make them an exceptionally useful non-human primate animal model for biomedical research in the fields of infectious diseases, oncology, autoimmune diseases, metabolic disorders, neuroscience, reproductive biology, stem cell research, and toxicology. Early stages of such research often include in vitro experiments which require standardized and well-characterized CM cell cultures derived from different tissues.

In this review we summarize the data on the main advantages, drawbacks and research applications of CM cell lines published to date including primary, immortalized, lymphoblastoid cell lines, embryonic stem cells and induced pluripotent stem cells (Table 1). Despite the popularity of CMs in biomedical research, the number of available standardized, well-defined, stable and validated CM cell lines is still small, and the use of CM cell lines obtained and characterized so far is often limited by the work of the laboratories that obtained the original cells. Thus, development of new standardized and commercially available CM cell lines will significantly increase the reproducibility and translational potential of experimental results obtained using CMs in many areas of biomedical research.

## Figures and Tables

**Figure 1 cells-12-02020-f001:**
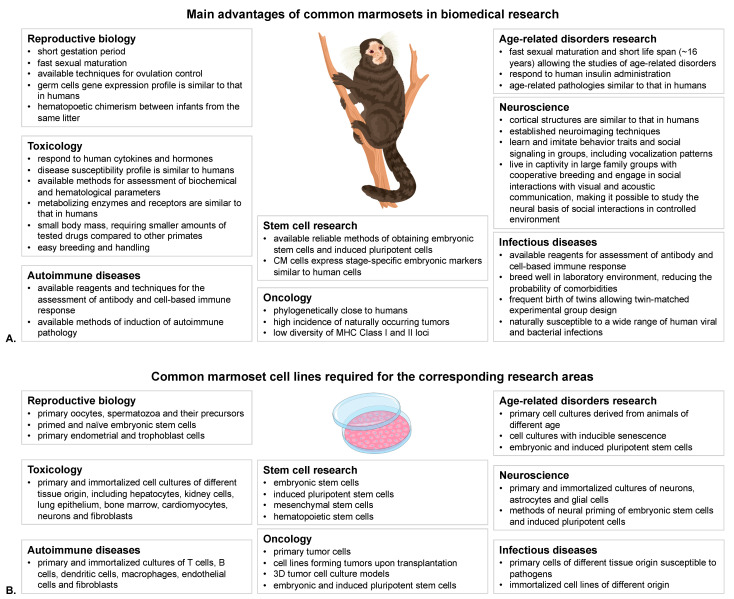
Main advantages of common marmosets in biomedical research (**A**) and cell lines required for the corresponding research areas (**B**).

**Table 1 cells-12-02020-t001:** Types of common marmoset cell lines used in biomedical research.

Type of CM Cell Lines	Advantages	Drawbacks	Cell Origin
Primary CM cells	easily obtainable; intact phenotype and unmodified genotype; sensitive to various pathogens	short culturing period; require repeated isolation of cells; high variability between batches	kidney cells [25]; skin fibroblasts [26,27,28]; hepatocytes [38,41,43,44,45,46]; mesenchymal stem cells [29,30,31]; dendritic cells [32]; peritubular cells [33] ovarian cells [34]
Immortalized CM cells	long culturing period; possible standardization; retain most properties of origin cells	lack of universal immortalization technique; possible differences in morphology, karyotype and gene expression pattern compared to primary cells	ovarian granulosa and theca cells [53]; skin and muscle fibroblasts [54,56,59]; hepatic progenitor cells [55]
Lymphoblastoid CM cells	reproducible transformation with both EBV and herpesviruses	safety issues due to production of viral particles in culture	EBV-transformed lymphoblastoid cells [61,62]; Herpesvirus saimiri-transformed lymphoblastoid cells [71,77]; Herpesvirus ateles-transformed lymphoblastoid cells [25]
CM embryonic stem cells	long culturing period and rapid proliferation with normal karyotype; express stage-specific embryonic markers similar to human cells; can be differentiated into cells of all three germ layers; no genetic manipulations required; possible auto-/allotransplantation without immunosuppression	the need for special equipment for obtaining; complicated culture conditions to maintain undifferentiated state; most published cell lines depend on mouse feeder cells, thus might be contaminated with mouse cell components; depend on fertilization, thus requiring a CM breeding colony; may form teratomas and malignant tumors upon transplantation	blastocyst-derived [81,83,85,98]; derived from early and compacted morula stage embryos [86]; cell lines established from both naturally-fertilized and in vitro fertilized embryos under both feeder and feeder-free conditions [101]
CM induced pluripotent stem cells	stable in culture; most reported cell lines exhibit normal karyotype; express pluripotency markers similar to human cells; can be differentiated into cells of all three germ layers; possibility to generate autological cell culture less prone to immunorejection	complicated pluripotency induction and differentiation protocols; possible high variability in the completeness of reprogramming; may form teratomas and malignant tumors upon transplantation due to possible tumorigenicity associated with reprogramming factors	iPSCs derived from: embryonic skin fibroblasts [113]; newborn marmoset fibroblasts [105,108,109,112]; adult skin fibroblasts [107,109,114]; fetal liver [109,111]; adult bone marrow-derived cells [118]

## Data Availability

Data sharing not applicable.

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
