# Peer review of "Common Marmoset Cell Lines and Their Applications in Biomedical Research"

_cells, 2023, doi:10.3390/cells12162020_

Round 1
Reviewer 1 Report
Manuscript entitled ”Common marmoset cell lines and their applications in biomedical research” by Bayurova E et al.
In this review the authors describe development and use of primary cultures and cell lines derived from common marmosets (CM). Experimentation with cells derived from CM are important because of close relation to Homo Sapiens and particularities of biology of the species. Although interesting in general, the topic is very specific and concerns only a small proportion of cell culture professionals. The manuscript contains a big number of factual mistakes that must be corrected by the authors.
Main issues:
1) Figure 1 describes how CM are used in biomedical research. This review is about cell cultures from CM, not the use of CM in science. Therefore, Figure 1 should be replaced by a figure describing the advantages and use of CM cultures.
2) Information of cell culture media and substrata should be added to the manuscript. Specifically, particularities of CM cell cultures that distinguish them from culturing of similar human cells should be discussed.
3) Lines 252 and 253. Human and mouse ES cells do NOT differ in mechanisms of self renewal (also, NOT cell renewal). The authors can read, for instance, article by Profs A Smith and J Nichols about different pluripotent states. Namely, what the authors call “human ES” represents primed pluripotency, while “mouse ES” represents naïve pluripotency. It is very well known that both human and mouse ES cells can be in both primed and naïve pluripotent state that is governed by growth factors/signalling molecules from the culture medium.
4) Table 1 is not consistent and contains factual mistakes. For instance, ES and iPS cells are very similar to each other and should contain similar list of Advantage and Disadvantages. The difference should be in a special equipment needed for development of ES cells and a possibility to generate autological cell culture for iPS cells. Importantly, even if ES cells are cultured on mouse fibroblasts used as substratum, they might NOT be contaminated with mouse genomic material. CM ES cells can be contaminated with mouse cells, but it is a completely different thing.
Minor issues:
1) Line 59. There is not a single ref cited by the authors that could support the exchange of ES cells between CM siblings. On the contrary ref 27 is clearly states that the exchange is limited to hematopoietic progenitors only. Therefore, phrase “embryonic stem cells” should be replaced by “hematopoietic progenitors”.
2) Lines 62-63. Allograft between species is called xenograft. The authors must use the right term.
3) Line 104. Word “affective” should be replaced by “effective”.
4) Lines 143-144. “In addition, calculation of population doubling time 143 showed that the derived hTERT-transgenic lines had significantly higher proliferation po-144 tential than the wild-type fibroblasts” is a repetition of previous phrase “expanded in vitro for 142 over 500 population doublings” and should be removed. Interestingly, the description of ref 53 by the authors repeats almost word for word the abstract of the article (with small reduction).
5) Line 172-174. This sentence is unclear and should be rewritten.
6) Lines 232-233. Phrase “strong differentiation activity” should be replaced by “three germ layer differentiation capacity”.
7) Line 238-239. Phrase “cell lines were further differentiated into cardiomyocytes [82] as preclinical model for allogenic transplantation after spinal cord injury [83]” should be rewritten, because cardiomyocytes can not be a preclinical model of spinal cord injury.
8) Line 274-275. The sentence is unclear and should be rewritten.
9) Line 297. Phrase “Yamanaka hypothesized that the factors that play important” is misleading, because the fact was proven in the article.
10) Line 301-302. Sentence “This work was awarded the Nobel 301 Prize in Physiology or Medicine in 2012” should be removed.
11) Line 319-320. Phrase “express high levels of NANOG, OCT4 and SOX2 mRNAs, have markers of pluripotency (nuclear Oct4, and plasma membrane TRA-1-81 and SSEA-4)” is misleading and should be rewritten, because NANOG, OCT4, SOX2, TRA-1-81 and SSEA-4 are all markers of pluripotency.
12) Line 352. Phrase “human dysgerminoma-like tumors” should be replaced by “tumors similar to human dysgerminoma”.
13) Line 375. “In the last years” should be replaced by “Recently”.
14) Line 378. “by Arravalli et al.” needs an actual reference to that article.
15) Line 393-394. “(Error! Reference source not found” should be removed.
Small corrections of grammar and, particularly, punctuation are needed.
Author Response
Reviewer 1: In this review the authors describe development and use of primary cultures and cell lines derived from common marmosets (CM). Experimentation with cells derived from CM are important because of close relation to Homo Sapiens and particularities of biology of the species. Although interesting in general, the topic is very specific and concerns only a small proportion of cell culture professionals. The manuscript contains a big number of factual mistakes that must be corrected by the authors
Response: Dear Reviewer, thank you very much for the thorough review and extremely important critical comments. Revision of the manuscript following these points helped us to considerably improve it. Although the topic is indeed very specific, the number of biomedical researches using Common marmosets (CM) is increasing annually. We hope that all the available information on CM cell cultures summarized in our work will be in demand among researchers.
Comment 1: Figure 1 describes how CM are used in biomedical research. This review is about cell cultures from CM, not the use of CM in science. Therefore, Figure 1 should be replaced by a figure describing the advantages and use of CM cultures
Response: During the process of manuscript preparation we were advised to add a figure describing the current use of CM in different areas of biomedical researches as not everyone is familiar with this laboratory animal model and its advantages above other primates and other small laboratory animals. Nevertheless, we agree with Reviewer 1 that this figure lacked information about cell lines which is the main focus of the review. We added information about the cell lines required for different areas of biomedical research to Figure 1 as an additional panel.
Comment 2: Information of cell culture media and substrata should be added to the manuscript. Specifically, particularities of CM cell cultures that distinguish them from culturing of similar human cells should be discussed
Response: We thank the Reviewer for the valuable comment. We have analyzed all the articles cited in the Review and found no specific features of CM cells culturing. Primary marmoset cells and immortalized cell lines could be cultured using MEM, DMEM or DMEM/F12 media supplemented with 10% FBS. For primary cells ITS+ supplement could be added. Lymphoblastoid cell lines cultured routinely using RPMI-1640 with 10% FBS. For ECS and iPSCs serum-free media with 20% serum substitute are used (KnockOut media). The only difference is that for CM feeder-free culturing is described only in a few recent articles and not widely used yet, however it is used for CM iPSCs. This difference was briefly mentioned in the Review. To make it clearer we added information about similarities in culturing through the manuscript. We believe that addition of specific culturing conditions to all the cell cultures listed in the Review will lead to its unnecessary complication. Additionally, during this round of revision we mentioned that reprogramming factors used to obtain CM iPSCs are of human origin due to high homology between human and CM corresponding genes and proteins. We added this information to the manuscript.
Comment 3: Lines 252 and 253. Human and mouse ES cells do NOT differ in mechanisms of self renewal (also, NOT cell renewal). The authors can read, for instance, article by Profs A Smith and J Nichols about different pluripotent states. Namely, what the authors call “human ES” represents primed pluripotency, while “mouse ES” represents naïve pluripotency. It is very well known that both human and mouse ES cells can be in both primed and naïve pluripotent state that is governed by growth factors/signalling molecules from the culture medium.
Response: While we completely agree with the Reviewer’s comment, the differences in pluripotent states between mouse and human cells were not the topic of this review, thus the sentence in question was removed.
Comment 4: Table 1 is not consistent and contains factual mistakes. For instance, ES and iPS cells are very similar to each other and should contain similar list of Advantage and Disadvantages. The difference should be in a special equipment needed for development of ES cells and a possibility to generate autological cell culture for iPS cells. Importantly, even if ES cells are cultured on mouse fibroblasts used as substratum, they might NOT be contaminated with mouse genomic material. CM ES cells can be contaminated with mouse cells, but it is a completely different thing.
Response: The text of the table was corrected according to the Reviewer’s recommendations
Comment 5: Line 59. There is not a single ref cited by the authors that could support the exchange of ES cells between CM siblings. On the contrary ref 27 is clearly states that the exchange is limited to hematopoietic progenitors only. Therefore, phrase “embryonic stem cells” should be replaced by “hematopoietic progenitors”.
Response: Thank you for the comment. The mechanism of hematopoietic chimerism in CMs was described in the previous sentence and the phrase “embryonic stem cells” was misleading. The text was corrected.
Comment 6: Lines 62-63. Allograft between species is called xenograft. The authors must use the right term.
Response: The text was corrected.
Comment 7: Line 104. Word “affective” should be replaced by “effective”.
Response: The text was corrected.
Comment 8: Lines 143-144. “In addition, calculation of population doubling time 143 showed that the derived hTERT-transgenic lines had significantly higher proliferation po-144 tential than the wild-type fibroblasts” is a repetition of previous phrase “expanded in vitro for 142 over 500 population doublings” and should be removed. Interestingly, the description of ref 53 by the authors repeats almost word for word the abstract of the article (with small reduction).
Response: Thank you for your comment. We’ve been doing CM cell lines immortalization with TERT for several years by now, read the article by Petkov et al. maybe a dozen times and have never noticed the repeat. The text was corrected.
Comment 9: Line 172-174. This sentence is unclear and should be rewritten.
Response: The sentence was rewritten
Comment 10: Lines 232-233. Phrase “strong differentiation activity” should be replaced by “three germ layer differentiation capacity”.
Response: The text was corrected
Comment 11: Line 238-239. Phrase “cell lines were further differentiated into cardiomyocytes [82] as preclinical model for allogenic transplantation after spinal cord injury [83]” should be rewritten, because cardiomyocytes can not be a preclinical model of spinal cord injury.
Response: A part of the sentence was lost during editing. The text was corrected.
Comment 12: Line 274-275. The sentence is unclear and should be rewritten.
Response: The sentence was rewritten
Comment 13: Line 297. Phrase “Yamanaka hypothesized that the factors that play important” is misleading, because the fact was proven in the article.
Response: The text was corrected
Comment 14: Line 301-302. Sentence “This work was awarded the Nobel 301 Prize in Physiology or Medicine in 2012” should be removed.
Response: The text was corrected
Comment 15: Line 319-320. Phrase “express high levels of NANOG, OCT4 and SOX2 mRNAs, have markers of pluripotency (nuclear Oct4, and plasma membrane TRA-1-81 and SSEA-4)” is misleading and should be rewritten, because NANOG, OCT4, SOX2, TRA-1-81 and SSEA-4 are all markers of pluripotency.
Response: Here we meant to separate the mRNA expression data from the immunostaining data, but the text was not clear enough. The text was rephrased
Comment 16: Line 352. Phrase “human dysgerminoma-like tumors” should be replaced by “tumors similar to human dysgerminoma”.
Response: The text was corrected
Comment 17: Line 375. “In the last years” should be replaced by “Recently”.
Response: The text was corrected
Comment 18: Line 378. “by Arravalli et al.” needs an actual reference to that article.
Response: Authors thank Reviewer for the comment. Required reference was added to the manuscript.
Comment 19: Line 393-394. “(Error! Reference source not found” should be removed.
Response: We do not see this error in our software, could be a link malfunction. The link to Table 1 was removed from lines 393-394
Reviewer 2 Report
Cell lines are an essential tool in biomedical research, allowing scientists to study and understand various diseases, develop new treatments, and test the effectiveness of drugs. Here the authors present a well laid out support for the use of marmoset cell lines in biomedical research.
Overall, I was left believing that,
1. The genetic and physiological similarities make marmoset cell lines particularly useful for studying human diseases and processes.
2. Cell lines from marmosets allow researchers to investigate the molecular and cellular basis of certain diseases that closely resemble those seen in humans.
Taken together, its easy to see why the authors believe CM cell lines offer an alternative approach that will improve reproducibility and translational potential of biomedical research.
I commend the authors for a comprehensive figure 1 and very clear table 1, which helped see the current uses of CM cell lines and the advantages and disadvantages clearly aligned.
While the manuscript needs some minor grammatical editing, I felt it read well and was logically structured.
Author Response
Dear Reviewer, thank you very much for the review and for high appreciation of our manuscript. We additionally improved Figure 1 and Table 1 according to the comments of Reviewer 1.
Reviewer 3 Report
The paper only described the collection of common marmoset ESCs and iPSCs. more specific perspective of these cell line and their possible use for biomedical research should be discussed.
English editing is needed.
Author Response
Dear Reviewer, thank you very much for the thorough review of our manuscript. Indeed, the purpose of this review is to summarize all of the available common marmoset (CM) cell lines, namely primary cells, immortalized cell cultures and ESCs, iPSCs among others. We’ve described the main features and perspectives or published use for the majority of cell lines in the review. Regarding more specific and detailed data on possible use for biomedical research of ESCs, iPSCs, as it was mentioned in text, this topic was excessively described in a recent review of Arivalli et al (10.3390/genes11070729) and is out of purpose of our review. We believe that with increasing interest of common marmoset as laboratory animals and a lack of commercially available CM cell lines, a review about different CM cell cultures obtained in the last fifty years is a valuable input to the scientific community.
Round 2
Reviewer 1 Report
The authors have addressed all my concerns.